# Rare Occurrence of *Blastocystis* in Pet Animals and Their Owners in the Pomeranian Voivodeship in Poland in the Light of Literature Data

**DOI:** 10.3390/jcm11112975

**Published:** 2022-05-25

**Authors:** Monika Rudzińska, Beata Kowalewska, Monika Kurpas, Beata Szostakowska

**Affiliations:** 1Department of Tropical Medicine and Epidemiology, Faculty of Health Sciences, Medical University of Gdansk, 80-210 Gdansk, Poland; mrudz@gumed.edu.pl (M.R.); bkowal@gumed.edu.pl (B.K.); 2Department of Immunobiology and Environmental Microbiology, Faculty of Health Sciences, Medical University of Gdansk, 80-210 Gdansk, Poland; monika.kurpas@gumed.edu.pl; 3Department of Tropical Parasitology, Faculty of Health Sciences, Medical University of Gdansk, 80-210 Gdansk, Poland

**Keywords:** *Blastocystis*, pet animals, humans, zoonotic transmission, Poland

## Abstract

*Blastocystis* is an intestinal microeukaryote with ambiguous pathogenicity, commonly detected in human feces worldwide. It comprises at least 28 genetically diverse subtypes (STs), 12 of which also occur in a wide range of animal species, giving rise to suspicion of zoonotic transmission. To investigate this, we conducted a molecular study of 145 stool samples of pet animals, and 67 of their owners, living in an urban area in Poland. *Blastocystis* was detected in only three (2.1%) animal samples (of two bearded agamas and a leopard gecko), while all dogs, cats, and pet rodents were *Blastocystis*-negative. *Blastocystis* was also present in three (4.5%) owners of animals, but they were cat owners, not reptile owners, and the subtypes identified in them differed significantly from those of reptiles. Additionally, the frequency of *Blastocystis* in different groups of dogs (depending on how they were kept) was analyzed. This work is the first to find *Blastocystis* in pet reptiles, and we encourage further investigation of *Blastocystis* in this poorly examined group of animals, as well as continued study on the transmission of this microorganism between humans and animals.

## 1. Introduction

*Blastocystis* is a common enteric protist harbored by humans and numerous animal species (mammals, birds, reptiles, amphibians, insects, and even oysters) [1,2,3]. As for humans, *Blastocystis* is the most frequently encountered eukaryotic microorganism in fecal specimens, worldwide, with a higher prevalence observed in countries with poor sanitary–hygienic conditions [4,5,6,7,8,9,10,11]. In spite of dozens of publications on the impact of *Blastocystis* on human health, its pathogenicity has not been established. Some reports show that *Blastocystis* can colonize human intestines for many years without causing any discomfort [12], while others suggest that *Blastocystis* may be a causative agent of intestinal and skin disorders (diarrhea, flatulence, abdominal pain, urticaria) of varying severity and recurrence rates [13,14,15,16,17,18,19,20,21]. The interaction between *Blastocystis* and irritable bowel syndrome (IBS) is also a subject of ongoing debate [22]. Although *Blastocystis* isolated from humans and animals are morphologically indistinguishable, in fact, both human and animal isolates show high genetic variability [23,24,25]. Based on sequence similarity within the small subunit ribosomal RNA gene (SSU rRNA), at least 28 subtypes (STs) of *Blastocystis* have been recognized, of which some show greater, while others lesser, host specificity [26,27,28].

Humans harbor twelve subtypes, i.e., ST1-ST10, ST12, and ST14; however, >90–95% of infections are caused by ST1-ST4, with a predominance of ST3 and ST1. ST2 and ST4 are observed slightly less frequently, with ST4 occurring mainly in Europe (in some European countries, it is even the dominant subtype). ST5-ST7 is found less frequently in humans, and the detection of ST8, ST9, ST10, ST12, and ST14 is reported in very few reports [4,11,29,30]. All subtypes have been reported in animals, which may suggest that animals could be a source of *Blastocystis* infection for humans. This seems to be confirmed by the few reports showing not only the presence of the same subtypes of *Blastocystis*, but also identical or very similar nucleotide sequences of *Blastocystis* isolated from animals and humans that are in close contact with each other [31,32,33,34,35,36,37]. It is known that pet animals, such as dogs and cats, can be the source of many parasitic infections for humans, e.g., *Toxocara canis*/*cati*, *Diphylidium caninum*, *Giardia intestinalis*, *Cryptosporidium* spp., *Sarcocystis* spp., or *Toxoplasma gondii*. Humans become infected orally through direct contact with animals, or through water, food, or an environment contaminated with animal feces containing developmental forms of parasites [38,39,40,41,42]. According to current knowledge, transmission of *Blastocystis* occurs in the same way [43,44,45,46]. Therefore, it is worth investigating whether pet animals can be a reservoir of *Blastocystis* infection for humans. The aim of this pilot study was to assess the prevalence of *Blastocystis* infection, and to compare *Blastocystis* genotypes isolated from pet animals and their owners to find out if there is transmission between them. Additionally, the available literature data on the occurrence of *Blastocystis* in pet animals was analyzed and discussed.

## 2. Materials and Methods

The analysis included 145 fecal samples collected from various pet animals, and 67 from their owners. The samples were collected from May 2018 to May 2020 in the Tri-City agglomeration (Pomeranian Voivodeship, Poland), which has about 800,000 inhabitants (data on 27 November 2021). The owners of the animals completed a questionnaire containing data on the breed, age, and living conditions of the animals. The animal group consisted mainly of dogs and cats (*n* = 64 and *n* = 54, respectively), as well as a small number of other animals such as bearded agamas (*n* = 6), rabbits (*n* = 4), guinea pigs (*n* = 4), leopard geckos (*n* = 3), rats (*n* = 3), monkeys of unknown race (*n* = 2), chinchillas (*n* = 2), degus (*n* = 1), a Syrian hamster (*n* = 1), and a Greek tortoise (*n* = 1).

Due to financial constraints, in the case of animals cared for by several family members, a stool sample was collected from the person who had the most contact with the animal. Not all pet owners provided their own stool samples for testing, hence the large difference between the size of the group of animals and the size of the group of owners. In a few cases, the same person owned several animals (e.g., six dogs, two dogs, two cats, a dog and a cat, three geckos). Only eight dogs were kept outside, and the remaining fifty-six dogs lived with their owners at home. Thirty-four cats were kept exclusively indoors, and twenty were allowed access to the outside. According to the respondents, all animals were well cared for, properly fed, and, in the event of health problems, under veterinary care. All dogs and cats were dewormed regularly.

The owners of the animals were given a brief instruction on how to collect the material (stool samples). Samples were taken immediately after defecation into plastic disposable containers with 70% ethanol, and then stored at 4–5 °C until DNA isolation.

The commercial Genomic Mini AX Stool Kit (A&A Biotechnology, Gdynia, Poland) was used to isolate genetic material from the obtained stool samples. The isolation procedure was carried out in accordance with the manufacturer’s instructions. The obtained genetic material was preserved at −20 °C, and then PCR was performed with pan-*Blastocystis* primers: RD5 (5′-ATCTGGTTGATCCTGCCCAGT-3′) and BhRDr (5′-GAGCTTTTTAACTGCAACAACG-3′), which enable the detection of all identified and new *Blastocystis* subtypes [47]. The composition of the reaction mixture was described in [48], briefly: 12.5 µL PCR Mix HGC Plus (A&A Biotechnology, Gdynia, Poland), 1 µL of each primer (concentration 10 µM), 2 µL of genomic DNA, supplemented with deionized water up to 25 µL. The time-temperature profile was as follows: initial denaturation 4 min at 94 °C, followed by 35 cycles of 15 s at 95 °C, 15 s at 60 °C, 30 s at 72 °C, and a final extension step of 5 min at 72 °C [48].

The obtained PCR products were sequenced in both directions at the Laboratory of Molecular Biology Techniques, Adam Mickiewicz University in Poznan (Poland), using the standard procedure with amplification primers. The obtained SSU-rRNA sequences were compared with 55 reference sequences representing the most common *Blastocystis* subtypes, i.e., ST1, ST2, ST3, ST4, ST4a, ST5, ST6, ST6a, ST7, ST8, ST9, ST10, ST12, ST13, ST14, ST15, ST16, and ST17, which were retrieved from the GenBank database. Sequences were trimmed and assembled using the MEGA X software. A population analysis was performed with the MEGA X software using the maximum likelihood method, with 1000 bootstraps to determine the exact position and identity of the *Blastocystis* isolates [49]. The identification of *Blastocystis* subtypes was additionally verified using the PubMLST open sequence typing database, available on the website (www.pubmlst.org/Blastocystis accessed on 25 August 2021) [50,51].

Additionally, studies assessing the occurrence of *Blastocystis* in dogs living in various housing conditions (domestic dogs, dogs from rural areas, those staying in shelters, and stray dogs) have been reviewed, and the results are summarized in Table 1.

Statistical analysis of these data was performed using the odds ratio. In addition, the prevalence of *Blastocystis* in dogs from different environments was compared using a chi-squared analysis (chi-Squared Calculator—socscistatistics.com accessed on 2 August 2021).

## 3. Results and Discussion

In total, stool samples of 145 animals and 67 of their owners were tested for the presence of *Blastocystis*. PCR products corresponding to *Blastocystis* were obtained in only 3 out of 145 (2.1%) animal samples, which was confirmed by sequencing. The positive samples were from two bearded agamas and one leopard gecko, whereas *Blastocystis* was not detected in any of the canine, feline, or other small mammal or turtle fecal samples.

Similarly, in animal owners, only three out of sixty-seven (4.5%) samples were positive for *Blastocystis*. Sequencing revealed that they were ST3, ST4, and ST7. All of these persons were cat owners.

The *Blastocystis* nucleotide sequences obtained from the reptiles in this study differed significantly from the human and avian *Blastocystis* reference sequences from GenBank (Figure 1). A similar pattern was seen in other reptile studies [1,37,80].

So far, turtles and snakes have been tested the most frequently among reptiles, and it is from these animals that the highest number of positive results was obtained [1,81,82,83,84]. Only one report mentions testing of two geckos and one bearded agama, in which no *Blastocystis* was detected [80]. The *Blastocystis* sequences that were isolated from the reptiles in our study are located on the same branch as the GenBank *Blastocystis* sequences from rabbits, common gundas, and rodents (Figure 1). When compared with reference sequences, they showed 99% similarity to the ST17 sequence (KC148208). ST17 is the second most often identified subtype in rodents (in Mexico [2], Libya [85], United Arab Emirates [80], and China [86,87]). Although reptiles are more and more eagerly and more often kept in homes as pet animals, the fact that they are carriers of different *Blastocystis* subtypes than those identified in humans suggests that the risk of transmission of the microorganism between them and their keepers is low. However, to confirm this assumption, further studies with more samples are needed. The *Blastocystis* sequences obtained from reptiles in our study were also largely (91% and 89%, respectively) consistent with the DQ186644 and DQ186647 *Blastocystis* sequences from GenBank derived from insects (cockroaches) [88] (Figure 1). Perhaps an explanation for the similarity of *Blastocystis* isolated from reptiles and cockroaches is the fact that these insects—which are likely to contribute to the transmission of *Blastocystis* [89]—are often food for reptiles. However, to clarify this issue, further studies of *Blastocystis* from both these host groups are necessary.

Most of the literature data on *Blastocystis* in pet animals relate to dogs, and, to a much lesser extent, cats. Table 1 presents data on the prevalence of *Blastocystis* in dogs, reported by authors from different countries, grouped according to the living conditions of the animals (dogs kept indoors, rural, shelter-resident, and stray dogs).

Statistical analysis using the non-parametric chi-squared test allowed for the comparison of the occurrence of *Blastocystis* in the above-mentioned groups of dogs. The comparison of these groups showed statistically significant differences (Table 2 and Appendix A).

This meta-analysis showed that, in shelter dogs, *Blastocystis* is detected statistically significantly more often than in indoor, rural, and stray dogs. In stray dogs, *Blastocystis* was significantly more frequent than in indoor and rural dogs. The analysis also showed that in dogs from rural areas, *Blastocystis* is statistically significantly less frequent than in the other three groups. The odds ratio showed that the chances of detecting *Blastocystis* in dogs kept in shelters are 28.81, 3.75, and 1.35 times higher, respectively, than in dogs from rural areas, indoor dogs, and stray dogs. The chance of detecting *Blastocystis* in stray dogs is 21.29 and 2.77 times higher, respectively, than in rural and indoor dogs.

*Blastocystis* was not detected in any canine or feline stool sample in our study. Similarly, *Blastocystis* infection has not been found in dogs in Spain [61], Greece [60], Japan [64], China [56], Australia [34,52], USA [69], or Brazil [54,71]. In a dozen or so other studies, *Blastocystis* in dogs was detected over a wide frequency range, from 1.3% to 70.8% (Table 1), and sometimes up to 100% of dogs were infected; however, these were small groups of three to five animals (data not included in the table) [35,70,90]. An analysis of the results of these studies from many countries shows that the prevalence of infections may be related to the animal’s welfare. Although infected and uninfected dogs were observed in each analyzed group, the most frequently infected were shelter dogs, followed by stray, indoor, and rural dogs. Transmission of *Blastocystis* occurs through water, food, or an environment contaminated with cysts, where dogs exhibit coprophagic behavior. This means that dogs kept in shelters, where several or even a dozen or more individuals live together in small cages or runs, more often come into contact with feces and surfaces contaminated with cysts than do dogs from other groups. This likely explains the high infection rate in shelter-resident dogs. All dogs in our study were well cared for, well-fed, walked only under the supervision of their owners, were regularly dewormed, and received veterinary care in the event of health problems. All these constituents could have had a protective effect against *Blastocystis* infection, hence why the infection was not detected.

According to the literature data, among cats, as in the case of dogs, the percentage of infected animals varies significantly, i.e., from 0 to 67.3%, and in small groups (consisting of three animals) it reached 100% [38,42,52,55,58,61,63,73,91,92,93]. Based on the available data, it seems that in cats—as in dogs—infection is favored by staying in a shelter; however, the available data are too sparse to be statistically assessed. It might also appear that cats that go outside are more likely to catch the infection, but in the present study, neither cats that did not leave the house nor those allowed to go outside were infected.

In the studies performed so far, nine *Blastocystis* subtypes have been identified in dogs, namely ST1–ST8 and ST10 (Table 1), while in cats, six subtypes have been identified, i.e., ST1–ST4, ST10, and ST14 [63,69,91,94]. The same subtypes have been detected in humans. Moreover, in dogs, as in humans, ST1-ST4 are found more often than the other subtypes, and in cats, ST1 and ST3 were observed slightly more often. This may suggest that dogs and cats serve as reservoirs of *Blastocystis* infection for humans, but the opposite direction of transmission is equally likely. It is noteworthy that the dog owners in our study, like their dogs, were not infected with *Blastocystis*, while among the cat owners, there were three infected (each with a different subtype: ST3, ST4, and ST7). This suggests they could not catch the infection from their own cats, as their own cats were negative for *Blastocystis.*

Of the reports on the occurrence of *Blastocystis* in dogs and cats, only a few have provided data on both these animals and related humans. In a small urban community in the Philippines, the same ST1-ST5 subtypes were found in dogs and humans (some of them were dog owners), but the proportion of subtypes in humans and dogs was different. ST3 was most prevalent in both humans and dogs, but was more than twice as common in humans as in dogs (41.4% vs. 17.4%). ST1 was the second most common subtype in humans, and the least frequent in dogs (22.6% vs. 4.3%). The opposite was true for ST5 and ST2, which were about three times more frequent in dogs than in humans (13% vs. 4.1%, and 8.7% vs. 3.1%, respectively). The most similar values in both host groups were found for ST4 (14.8% in humans and 13% in dogs) [58]. Interestingly, ST1, ST2, ST3, ST4, and ST7, which were detected in people with gastrointestinal symptoms, were also present in their dogs and cats [35,67,90], and the sequence of *Blastocystis* isolated from a human in Thailand was 100% consistent with the sequence of an isolate derived from a dog from the same community [70].

*Blastocystis* was not detected in any of the rabbit and rodent samples in our study. As for rabbits, a few authors have observed a low percentage of animals infected with ST4 (1% and 3%), and a single case of ST14 infection [77,80,95]. In rodents, ten subtypes of *Blastocystis* (ST1-ST5, ST7, ST8, ST10, ST13, ST17) have been reported with a marked predominance of ST4, especially in rats, with infection rates varying from 0% to 100% [63,77,87,96,97,98,99,100,101]. There were only 11 rodents in our study, and it may be that *Blastocystis* was not detected in them due to the small number of animals tested. Since rodents are quite popular pet animals and, as shown in the literature, they appear to be the main animal reservoir of ST4 [87], a subtype that has been reported as a possible cause of bothersome diarrhea in humans [16,102], research on *Blastocystis* should be continued in these animals.

To sum up, our study is likely the first to detect *Blastocystis* in the bearded agama and leopard gecko. The genotypes of *Blastocystis* identified in these animals are significantly different from those found in humans; therefore, it seems that they are unlikely to infect humans. However, given that infected reptiles accounted for 30% of the small number of reptiles included in the study, and that these animals are increasingly being bred as pet animals, and because there is little knowledge of ‘reptilian’ subtypes, research on *Blastocystis* in this group of animals should be continued.

The reason that no *Blastocystis* was detected in any of the other animals may be that all samples were obtained from healthy and well-cared for animals. The failure to detect *Blastocystis* in any of the rabbits and rodents could have been due to the small number of animals sampled.

The results of our study indicate that further studies of *Blastocystis* isolated from animals and their caretakers are necessary to fully understand whether there is a transmission of these microorganisms between animals and humans, and how often.

## Figures and Tables

**Figure 1 jcm-11-02975-f001:**
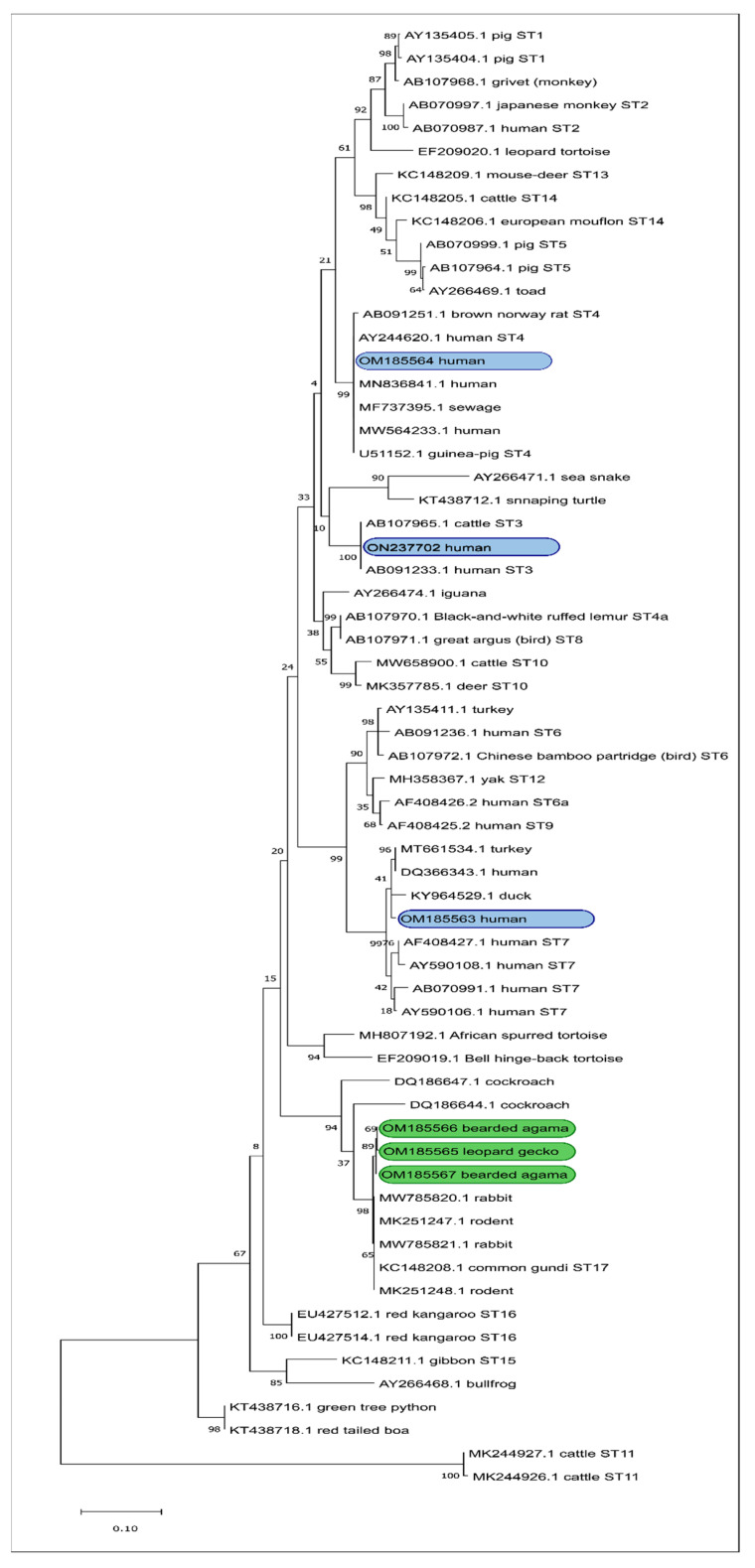
Molecular phylogenetic relationship of *Blastocystis* isolates performed using the maximum likelihood method with 1000 replications. Reptile isolates from this study are marked in green; human isolates are marked in blue.

**Table 1 jcm-11-02975-t001:** Infection rates and subtypes of *Blastocystis* identified in dogs from different countries, grouped according to the living conditions of the animals.

Country	Number of Animals Tested	Number of Infected Animals	(%)	Identified Subtypes (*n*)	References
**Indoor dogs**					
Australia	11	0	-	-	[52]
Australia	80	2	2.5	ST1 (2)	[53]
Brazil	20	0	-	-	[54]
Brazil	78	2	2.6	N/A	[55]
China	199	0	-	-	[56]
China	315	6	1.9	ST1, ST2 (the number of dogs tested was not reported)	[57]
Philiphines	145	20	13.8	ST2 (1), ST3 (2) ST4 (2), ST5 (2) ST1/3 (1), ST2/3 (1) ST4/5 (1), STnz (10)	[58]
France	116	4	3.4	ST2 (2), ST10 (2)	[59]
Greece	30	0	-	-	[60]
Spain	55	0	-	-	[61]
Iran	120	22	18.3	N/A	[62]
Iran	154	29	18.8	ST2 (8), ST3 (11), ST4 (3), ST7 (3), ST8 (2), ST10 (2)	[63]
Japan	27	0	-	-	[64]
Colombia	175	32	18.3	N/A	[65]
Colombia	8	1	12.5	ST1 (1)	[66]
Poland	31	1	3.2	ST7	[67]
Thailand	13	1	7.7	ST3	[68]
USA	51	0	-	-	[69]
**Dogs from rural areas ***					
Australia	5	0	-	-	[70]
Australia	45	0	-	-	[52]
Brazil	11	0	-	-	[71]
Brazil	38	0	-	-	As above
Chile	30	1	3	N/A	[72]
Cambodia	80	1	1.3	ST2	[53]
**Shelter dogs**					
Australia	72	51	70.8	N/A	[73]
Australia	5	0	-	-	[70]
China	149	35	5.4	ST1 (6), ST3 (28) ST10 (1)	[56]
USA	103	10	9.7	ST1, ST10	[69]
Italy	99	21	21.2	ST3	[74]
**Stray dogs**					
Greece	42	0	-	-	[60]
India	80	19	23.8	ST1 (9), ST4 (2) ST5 (1), ST6 (7)	[53]
Iran	181	37	20.4	N/A	[62]
Japan	27	0	-	N/A	[64]
**Other dogs ****					
Australia (various dogs)	300	10	3	N/A	[75]
Australia (various dogs)	10	2	20	ST1 (1), STnz (1)	[70]
Argentina (no data)	139	4	2.9	N/A	[76]
China (various dogs)	136	4	2.9	ST1 (3), ST4 (1)	[77]
Turkey (kept indoors and from rural area)	200	0	-	-	[78]
China (from breeders)	237	0	-	-	[56]
China (from the market)	60	0	-	-	As above
Chile (with diarrhea)	972	351	36.1	N/A	[38]
Colombia (no data)	40	15	37.5	ST2	[79]

* The reports did not specify whether the dogs were kept outside or lived with their owners at home. ** dogs did not qualify for any of the above groups due to lack of or incomplete data on living conditions and number of animals. N/A—not applicable; the study was performed using a microscope or microscope and in vitro culture. STnz—subtype not identified.

**Table 2 jcm-11-02975-t002:** The *p*-values for the chi-squared analysis comparing the prevalence of *Blastocystis* in dogs kept under different conditions, calculated on the basis of the data from Table 1 and Appendix A.

Dogs	Kept Indoors	Rural	Shelter-Resident	Stray
Kept indoors	1	0.000455	<0.00001	<0.00001
Rural	0.000455	1	<0.00001	<0.00001
Shelter-resident	<0.00001	<0.00001	1	0.00054
Stray	<0.00001	<0.00001	0.00054	1

## Data Availability

The sequences obtained and analyzed during this study were deposited in the GenBank database under the accession numbers OM185563–OM185567, ON237702.

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
