# Peer review of "Rare Occurrence of Blastocystis in Pet Animals and Their Owners in the Pomeranian Voivodeship in Poland in the Light of Literature Data"

_jcm, 2022, doi:10.3390/jcm11112975_

Round 1

Reviewer 1 Report

The manuscript titled “Rare occurrence of Blastocystis in pet animals and their owners in the Pomeranian Voivodeship in Poland in the light of literature data” has been reviewed for Journal of Clinical Medicine. The manuscript is written in effective way. The methodology is quite up to standards. The results are presented in interesting manner. However, in discussion part the references should needs more update especially from year 2021.

Author Response

Thank you for your valuable comments that have allowed us to improve the manuscript

We re-examined the recent literature on Blastocystis in pet animals and pet animals owners and we've added the new reference from 2021 (line 229, ref. 95).

Reviewer 2 Report

The authors wonder about the transmission of blastocystis between pets and their owners. The method is well run. Their results differ from those described in the literature: they did not find blastocystis in dogs and cats. They discuss it very well. On the other hand, in an interesting way, they find it in reptiles. As pointed out by the authors, further studies are needed to confirm this fact.
Somes détails can be improved

Line 96:the reference does not detail the PCR and refers to another reference. Please briefly describe the PCR protocol, revelation method included
Line 130:In animal owners with positive detection of blastocystis, please detail which animals they own

Author Response

Thank you for your valuable comments that have allowed us to improve the manuscript.

We have changed the reference concerning PCR conditions from:

Rudzińska, M.; Kowalewska, B.; Waleron, M.; Kalicki, M.; Sikorska, K.; Szostakowska, B. Molecular Characterization of Blastocystis from Animals and Their Caregivers at the Gdańsk Zoo (Poland) and the Assessment of Zoonotic Transmission. Biology (Basel). 2021, 10, doi:10.3390/biology10100984.

to:

Rudzińska, M.; Kowalewska, B.; Szostakowska, B.; Grzybek, M.; Sikorska, K.; Świątalska, A. First report on the occurrence and subtypes of Blastocystis in pigs in Poland using sequence-tagged-site pcr and barcode region sequencing. Pathogens 2020, 9, 595.

Additionally we have briefly described PCR mixture and reaction conditons (lines 106-111, ref. 48).

Missing information regarding infected persons, i.e. that all of them were cat owners, has been added (lines 148-149).

Reviewer 3 Report

The manuscript by Rudzińska et al. is well done and the results are significant. I only have minor comments on public health importance of Blastosystis as follows:

In the introduction or discussion, the authors should briefly mention complications induced by Blastosystis hominis (e.g. irritable bowel syndrome) in humans. For this, you can refer to a recently published study:  “The neglected role of Blastocystis sp. and Giardia lamblia in development of irritable bowel syndrome: A systematic review and meta-analysis. https://www.sciencedirect.com/science/article/pii/S0882401021004897”

Author Response

Thank you for your valuable comments that have allowed us to improve the manuscript.

According to the Reviewer suggestion, in Introduction we added an information about the interaction between Blastocystis and irritable bowel syndrome (IBS) with suggested reference (lines 43-44, ref. 22).